# Long Bone Defect Filling with Bioactive Degradable 3D-Implant: Experimental Study

**DOI:** 10.3390/biomimetics8020138

**Published:** 2023-03-28

**Authors:** Arnold Popkov, Natalia Kononovich, Gleb Dubinenko, Elena Gorbach, Alexander Shastov, Sergei Tverdokhlebov, Dmitry Popkov

**Affiliations:** 1Ilizarov National Medical Research Center for Traumatology and Orthopedics, Kurgan 640014, Russia; popkov72@gmail.com (A.P.); n.a.kononovich@mail.ru (N.K.); gorbach.e@mail.ru (E.G.); alshastov@yandex.ru (A.S.); 2School of Nuclear Science & Engineering, National Research Tomsk Polytechnic University, Tomsk 634050, Russia; dubinenko.gleb@gmail.com (G.D.); tverd@tpu.ru (S.T.)

**Keywords:** bone defect, degradable implant, polycaprolactone, hydroxyapatite, osseointegration

## Abstract

Previously, 3D-printed bone grafts made of titanium alloy with bioactive coating has shown great potential for the restoration of bone defects. Implanted into a medullary canal titanium graft with cellular structure demonstrated stimulation of the reparative osteogenesis and successful osseointegration of the graft into a single bone-implant block. The purpose of this study was to investigate osseointegration of a 3D-printed degradable polymeric implant with cellular structure as preclinical testing of a new technique for bone defect restoration. During an experimental study in sheep, a 20 mm-long segmental tibial defect was filled with an original cylindrical implant with cellular structure made of polycaprolactone coated with hydroxyapatite. X-ray radiographs demonstrated reparative bone regeneration from the periosteum lying on the periphery of cylindrical implant to its center in a week after the surgery. Cellular structure of the implant was fully filled with newly-formed bone tissue on the 4th week after the surgery. The bone tissue regeneration from the proximal and distal bone fragments was evident on 3rd week. This provides insight into the use of bioactive degradable implants for the restoration of segmental bone defects. Degradable implant with bioactive coating implanted into a long bone segmental defect provides stimulation of reparative osteogenesis and osseointegration into the single implant-bone block.

## 1. Introduction

Unsatisfactory outcomes of orthopedic surgery for non-unions and bone defects are not rare [1,2,3]. The methods of transosseous osteosynthesis developed by G. Ilizarov based on low-invasive surgery, fixation stability, preservation of blood supply, and functional loading demonstrated its efficiency in the treatment of pseudarthrosis and bone defects [4,5,6]. On the other hand, inconveniencies and disadvantages of external fixation are numerous: long-lasting external frame wearing, staged in-patient stay, delayed functional recovery, high rate of pin-site infections, decreased life quality of patients with external frame, and external fixation index in single-level lengthening for bone transport on tibia being over 40 days/cm in adult patients [7,8,9]. Recently, an idea appeared to fill bone defects with a personalized titanium implants, the design of which (inner cells and pores) is similar to the structure of spongy bone [10,11,12]. Experimental studies demonstrated that osseointegration of such an implant provides fast and durable restoration of a bone defect, but the titanium does not allow monitoring the process of reparative bone regeneration and its functional restructuring in clinical conditions by the methods of X-rays [13,14,15]. Furthermore, sometimes a metal implant remained into the bone arises concerns of a patient.

Over the past few years, there has been significant interest in the use of artificial bioresorbable scaffolds made by 3D printing to treat large bone defects [16,17,18]. One of the main advantages of 3D printing is the ability to personalize the scaffold, which can be designed to fit the size and shape of the bone defect with a similar structure and morphology to the tissue it will replace. Fused Deposition Modeling (FDM) is a simple and cost-effective 3D printing technique that allows for printing with thermoplastic polymers and polymer composites with a print resolution of up to 50 μm. Currently, new linear polyester-based bioresorbable materials for FDM 3D printing of personalized scaffolds for bone tissue engineering are actively studied. Polylactic acid (PLA) and PLA-based composites have been the most extensively studied, as they show good printability, a high level of biocompatibility, support cell adhesion and proliferation, and can accelerate the formation of bone tissue and osseointegration at the defect site in the combination with calcium phosphates [19,20,21,22]. An alternative to PLA is another linear polyester polycaprolactone (PCL). Compared to PLA, PCL has a much lower glass transition temperature, unique elastic properties at human body temperature, and is known for its good suturability [23]. The low melting temperature of PCL allows for adapting the strategy of FDM printing, and its high elasticity makes it easy to implant and withstand cyclic loads [24,25]. Previously, it was shown that 3D-printed PCL composite scaffolds filled with hydroxyapatite (HA) demonstrated a good level of biocompatibility, support for cell adhesion, growth, and proliferation [26,27].

Recent studies have shown that the printed infill pattern is also important for the biocompatibility of biomimetic scaffolds [28,29,30]. Triply periodic minimal surfaces (TPMS) have gained attention due to their high surface-to-volume ratio and interconnected porosity, which closely mimic the structure of natural materials [31]. Numerous of studies have employed gyroid-like TPMS infill patterns for 3D printing of PCL-based scaffolds and showed improved cytocompatibility, mechanical strength, and osteoblast proliferation compared to conventional cubic and rectangular infill patterns [32,33,34,35,36,37,38]. Studies demonstrate the potential of using TPMS structures in the design and fabrication of personalized biomimetic scaffolds for bone tissue engineering.

The purpose of this experimental study was the assessment of the reparative osteogenesis and osseointegration of a biomimetic TPMS-structured degradable cylindrical implant with bioactive coating for long bone defect filling.

## 2. Materials and Methods

This non-randomized study was performed on five mixed-breed sheep of feminine sex aged of 1 year in 2022. All the cheeps received the same postoperative care and same complete feeding, they were examined by veterinarian doctor 5 days per week. They were allowed to walk with no limitations in adapted vivarium.

Applied implants for bone defect filling had dimensions of 40 mm height and 18 mm diameter (Figure 1a,b). The implants were 3D printed of ε-polycaprolactone (PCL; Sigma-Aldrich, St. Louis, MO, USA; Mn 80,000 g/mol) and coated with hydroxyapatite (HA; Fluidinova, Portugal; particle size 10 ± 5 μm). The implants were fabricated via Fused Deposition Modeling (FDM) additive technology at Tomsk Polytechnic University (Tomsk, Russia). PCL pellets were melted and extruded with the use of Filabot EX2 (Filabot HQ, Barre, VT, USA) commercial single screw extruder to fabricate filament of 2.8 ± 0.15 mm diameter. The temperature of extrusion was 80 ± 2 °C and the rate of extrusion was 2 m/min. Extruded filament was used for the 3D printing of scaffolds with the use of Ultimaker S5 (Ultimaker B.V., Utrecht, The Netherlands) commercial FDM 3D printer. The temperature of the glass substrate was 35 °C, and the temperature of the printing nozzle was 200 °C. The printing was performed at printing rate of 6 mm/s. Dimensions and shape of implants were designed exactly in accordance with the transverse dimensions of the experimental animals tibial shaft and presented in Appendix A. The middle part of the implants was designed in the form of 20 mm long porous cylinder with 18 mm diameter and intramedullary cannel of 5 mm diameter. The inner space of the middle part of implants was printed in the form of a gyroid divided into cells of 1 mm diameter with walls thickness of 450 μm. Intramedullary canal of 5 mm diameter was designed in the center of the porous part. The outer walls of the cylinder had pores of 1 mm diameter to facilitate ingrowth of blood vessels to the bone defect from the side of periosteal tissues and intramedullary canal of adjacent bone fragments. The end face surfaces of the middle porous part were printed in the form of hollow cylinders of 10 mm long each with small pores of 1 mm. The external and internal surfaces of the implant were coated with a bioactive layer of HA. «Solvent/non-solvent» treatment of scaffolds was performed with the use of the mixture of toluene and ethanol at 3:7 *v/v* ratio. HA was mixed with the toluene/ethanol mixture at 10% *w*/*w* and stirred on magnetic stirrer for 30 min to obtain suspension. Scaffolds were dipped into the suspension for 2 min at room temperature under continuous stirring. Coated scaffolds were washed with ethanol and dried for 24 h under vacuum (1 mbar) at room temperature.

Protocol. The study protocol included the following phases:1–2 days before surgery: physical examination and standard AP and lateral radiographs of both tibiae including adjacent joints;D0: surgery;Postoperative period: standard AP and lateral radiographs were taken on the D1 and later once every week;Euthanasia at D30 (Euthanasia was performed by an intravenous injection of sodium thiopental 5%, 45 mg/kg).

Surgery. All the surgical interventions were performed under general anesthesia with a sodium thiopental 5% solution (10–15 mg/kg) with atropine as needed. The same surgeon performed all the surgeries. The first stage of the surgery consisted of the modeling of a tibial defect: a longitudinal incision was made along the anterior surface of the tibia and the shaft of the tibia was exposed. The periosteum was dissected longitudinally, and 20 mm long sector of the shaft was resected with an oscillating saw. Then, the cylindrical implant was mounted on the defect site by the integration of free ends of the bone fragments into hollow cylinders on the end face surfaces of the implant. The wound was sutured layer-by-layer, and the limb was fixed with the Ilizarov fixator. Regarding geometry of circular external fixator, the Ilizarov frame comprised two distal circular rings of 110 mm diameter made of stainless steel and two proximal rings of 110 mm diameter in three quarters connected by three 12 cm long threaded rods (Figure 2b). The gaps of proximal rings were oriented posteriorly as so allowing flexion motion in knee joint while walking. Three stain steel 1.5-mm wires were inserted in proximal tibial fragment and three Kirscher wires without olive of the same diameter were inserted in the distal tibial fragment (one of distal wires was inserted through tibia and fibula). Angulation between wires was about 90°. All wires were tensioned with 100 kg forces with graduated wire—tensioner. The external frame fixation lasted 30 days in each animal. In contrast to rigid half-pins, an external fixator with wires provides biomechanical benefit with the weight bearing on the bone regenerate while walking and does not demonstrate stiffness with negative influence on bone remodelling [39,40]. Cefazolin was administered intramuscularly 7 days after the surgery (0.5 g twice a day). In postoperative period the wounds underwent daily control, pin-site care was performed every 2 days. The experimental surgeries were performed by one surgical team.

Radiographically control was performed using Premium VET X-ray system (TOSHIBA (Rotanode) Model E7239. N: 10G749, Japan) and digital radiography system with CANON CXDI-401C COMPACT flat-panel detector (Canon Inc. Medical Equipment Group, Tochigi, Japan). The optical density of the interfragmentary gap filled with implant and newly formed bone tissue was measured with original software (Certificate for the Computer Program No. 2014611777 RF) by the transverse and longitudinal scanning (Figure 2a).

Computer tomography (CT) of anatomical specimens was preformed using the PRODIS Compact table microtomography system (PRODIS. NDT Ltd., Moscow, Russia) and VG Studio 3.5 (Volume Graphics GmbH, Heidelberg, Germany) software.

For morphological study the specimens were dissected in the longitudinal direction to visually assess the state of the bone and soft tissues lying on the implant surface, and to assess the tissue substrate filling the cells of the implant and the longitudinal axial canal of the implant.

AtteStat 12.0.5 (Kurgan, Russia) software was used for the statistical analysis. The statistical values described the mean and standard deviations.

Ethics. Approval was obtained from the institutional ethics committee before the experiment (Ethics Committee of the Ilizarov National Medical Research Center for Traumatology and Orthopedics, Kurgan, Russia (protocol code 1(71), date of approval 28 April 2022). Interventions, animal care, and euthanasia conformed to the requirements of the European Convention for the Protection of Vertebrate Animals used for Experimental and other Scientific Purposes (Strasbourg, 18 March 1986), principles of laboratory animal care (NIH publication number 85-23, revised 1985), and the national laws.

## 3. Results

Three-dimensional-printed PCL implant and CT of porous part of the implant are presented in Figure 1b,c, respectively. Implants have smooth surface with open pores in the middle part (Figure 1b). CT images show regular porous structure of the middle part of implant with visible inner intramedullary canal (Figure 1c).

A promising approach for modifying the surface of biodegradable polyesters involves treatment of the polymer with a mixture of organic solvents to partially swell its surface [41,42,43]. It was shown that a mixture of toluene and ethanol has no destructive effect on the PCL scaffold structure and allows non-covalent immobilization of gelatin and iodine. The swelled surface layer of the polymer is able to adsorb molecules and particles from the contacting medium. Such mechanism of adsorption is highly suitable for the immobilization of inorganic particles. HA distributed evenly over the surface of the implant in the form of dispersed segregated particles (white particles in the SEM micrograph of implant surface) (Figure 3). EDX mapping of the HA coated surface and chemical composition of implants examined by FTIR are shown in the supporting information manuscript (Appendix A).

The body weight of animals at the beginning of the study was 37.2 ± 3.2 kg (ranged from 34 to 38.5 kg). In a month after surgery the mean weight was 37.1 kg (34.5 kg to 37.9 kg). The sheep started walking with total weight-bearing on the limb that has undergone surgery since the 3th–5th day (Figure 2b). There was neither septic nor neurological complications during the study. The weight-bearing function of the experimental limb was maintained until the end of the study.

The radiography demonstrated to be enough informative for assessment of bone reparative regeneration: the PCL implant was X-ray transparent enough. X-ray transparency of the implant allowed monitoring of reparative bone regeneration and bone tissue ingrowth. On the first day of the study the cellular structure of the implant was clearly visible (Figure 4a). The optical density inside the cells corresponded to the density of the soft tissues surrounding the bone. Less pronounced shadow was visible along the axial channel of the implant, as well as the denser shadow from the superposition of muscle tissue.

A slight periosteal reaction was noted closing to the ends of the implant at the 7th day after the surgery (Figure 4b). Densitometry revealed that all the cells of the implant were filled with the substance which optical density is higher than the density of surrounding soft tissues. An area of a low density was located in the middle of the implant along the axial line. We hypothesized that this area was not yet filled with newly forming bone tissue of low mineralization. By the 21st day of the study, the optical density in transverse scanning increased. It was no longer possible to distinguish the area of the axial canal of the implant (Figure 4c).

We compared optical density of cylindrical implant in longitudinal scanning with the optical density of the distal tibial fragment (Table 1). The optical density of the implant was increasing during the study and approaching to the values of the optical density of the distal tibial fragment.

Longitudinal scanning of the implant zone compared to the density of the distal tibial fragment revealed increasing optical density during experimental study (Figure 5).

There was no pathological reaction from the soft tissues and signs of inflammation: subcutaneous tissue, fascia, muscles, and periosteum were of the usual structure. On the transverse and longitudinal sections of the implants after euthanasia (one month after the surgery), all cells of implants and longitudinal axial canals were filled with bone spongy substance (Figure 6a,b). Comparative analysis of the images with the use of CT reconstruction in the range of the shades of gray and color visualization revealed the presence of colored structures of high-intensity on the surface (in the outer pores of implants) and in the cells of the implant proving bone ingrowth (Figure 6c,d). In the longitudinal axial canal, images of trabecular structure of 5–6 mm long were observed (Figure 6c,e, white arrows). They were united with the proximal and distal bone fragments. The intensity of these structures corresponded to the intensity of cortical bone.

On the sides of the proximal and distal bone fragments, the implants were periosteally covered with cone-shaped bands of dense compact bone tissue (Figure 6e). Moreover, low mineralized bone trabeculae were found in the cells of implants (Figure 6d). There were no signs of resorption or porosity of the compact plate of a bone fragments.

Thus, the study demonstrated ingrowth of bone tissue into cylindrical PCL-HA implant both from the adjacent bone ends and periosteal structures. By the 30th postoperative day the implant and bone defect were filled in with a newly-formed bone tissue.

## 4. Discussion

Architecture and surface properties are among the most important characteristics for the biocompatibility of scaffolds. The ideal scaffold should provide an optimal surface for cell adhesion and interconnected porous architecture for vascularization and extracellular matrix reproduction [44,45]. Three-dimensional printing technologies implement a highly adaptable and predictable scaffold design: layer-by-layer fabrication allows the formation of various shapes and a wide range of infills from ordinary 2D grids to 3D structures with triply periodic minimal surfaces. Providing an interconnected 3D porous structure, high surface area and high-energy adsorption [46,47,48], TPMS appears to be biomimetic and highly desirable for the design of scaffolds. However, among various biomimetic TPMS structures, a gyroid is the most convenient to use due to the presence of the gyroid infill in advanced slicer software. The possibility to design the gyroid infill directly with the use of slicing software makes the scaffolds design highly variable and reproducible.

Ilizarov’s distraction osteogenesis method demonstrates effectiveness in treatment and reconstruction of large bone defects [3,4,5]. However, difficulties and complications associated with the use of the Ilizarov bone transport technique have been reported: high rate of pin-site infection, prolonged time of external frame fixation, joint contractures, risk of ischemic regeneration in compromised soft tissues around a large defect, and physical and psychological suffering of patients [4,5,49,50].

Combination of Masquelet technique (induced membrane technique) followed by Ilizarov bone transport demonstrated effectiveness in the treatment of nonunions of long bones [51,52]. The bone transport through induced membrane chamber allowed improving outcomes and avoiding bone grafting. However, the healing index remained high, varying from 35 to 60 days/cm [53].

Previously, it was demonstrated that an ideal type of non-free bone grafting for bone defects filling presents a vascularized autologous osteotomized bone fragment transported gradually in the interfragmental gap within the soft tissue envelopment providing osteogenic properties [54,55]. An autologous graft allows to avoid graft harvesting but risks of delayed bone union, pin-site infection, and prolonged external frame wearing are not negligible.

Another option for treatment of extensive diaphyseal and metaphyseal bone defects, emerging for the latest 20 years, is presented by 3D printed scaffolds for hard tissue replacement [56,57]. This technique gives an alternative method to repair bone defects caused by osteomyelitis or trauma [58,59]. Materials for large bone defect reconstruction required to ensure high resistance to mechanical load. The titanium porous implants corresponded to these requirements providing high mechanical resistance to loading force [60]. On the other hand, biomaterials used in bone defect repairing should provide osteoinductivity. It is the second requirement of an ideal bone graft material for total long bone defect reconstruction. A scaffold promoting new bone tissue formation should enable conducting of bone regeneration [61]. Non-degradable materials provide only maintenance of the anatomical function rather than bone regeneration. High stiffness of metallic biomaterials results in a stress-shielding effect leading to microfractures of additional bone tissue surrounding implants [62]. Habibovic et al. [63] and Lan Levengood et al. [64] advocate necessity of an intrinsic microporosity and macropores to enhance new bone formation on the surfaces of 3D scaffold implants. Moreover, the increase in the porosity of implants decreases the stress-shielding effect. However, the strength of implants simultaneously reduces [65].

Biomaterials used in bone defect repairing should provide osteoinductivity. Three-dimensional printing of calcium-phosphates scaffolds enables biocompatibility and osteoinductive properties of implants [66]. Depending on filling the 3D-printed titanium scaffolds can provide antibacterial and osteoconductive properties [67]. However, it is still limited compared to the one of rhBMP-2 regarding osteoinduction properties. The recombinant human BMP-2 stimulates mesenchymal stem cells to differentiate into preosteoblasts and osteoblasts, and acts as a trigger for migration of osteoblasts in vivo and in vitro [68]. The use of BMP-2 for clinical purposes arises concerns: the release of rh-BMP-2, especially in surrounding soft tissues, could cause heterotopic ossifications, osteolysis and cancer [69,70]. For this reason, an osteoinductive activity of an implant has to be limited to contacting surfaces of a reconstructed bone and implant to avoid any systemic undesirable side effect.

Coating of surfaces with hydroxyapatite (HA) improves bioactivity and osteointegration of metallic implants [71]. Only implants with bioactive HA-coated surfaces ensure primary implant stability which results in an improved contact between bone and implant [72]. The osteoinductive layer of HA on the implant can be achieved by micro arc oxidation in an electrolyte with calcium and phosphate or by composite coating including fluorocarbon plastic filled with dispersed hydroxyapatite [73]. Uklejewski et al., emphasize the role of a biomineral coating with a native Ca/P ratio [74]. It can be one of crucial factors in mechanics of interacting biomaterials the innovative multi-spiked connecting scaffold [75].

On the other hand, there are certain limitations of clinical use of titanium implants in long bone defects in humans. Porous 3D scaffolds fabricated of titanium alloys induce osseointegration but material obstructs radiological control of ingrowth of newly formed bone in early postoperative period and in long-term follow up after surgery in clinics [76].

Recently, a scaffold-guided bone regeneration with use of highly porous biodegradable scaffolds providing long-term mechanical stability has been emerged [76]. Our earlier morphological studies demonstrated a positive effect of implant hydroxyapatite coating on the reparative bone formation in experimental fractures: porous surfaces contribute to a microvascular development followed by formation of dense trabecular structure on the implant surface without formation of a connective tissue envelope [77]. In cases of a significant defect in long bone shaft and the absence of bone marrow and endosteum, the periosteum ensures the main role in the reparative osteogenesis process of restoring the bone substance [13]. The formation of bone tissue is associated primarily with the intensity of angiogenesis: perivascular cells that have differentiation potential similar to mesenchymal stem/progenitor cells migrate together with the vessels into the cells of the graft differentiating into one or more of the cellular components of the vascular bed [78,79]. Possessing multipotent properties and inducibility to osteogenesis, in the presence of HA (inducer), which is present on the surface of both external and internal structures of the new implant, perivasculitis differentiate into osteoblasts [73,80].

The proposed PCL-HA cylindrical implant ensures ingrowth of bone tissue into cells of implant from the surrounding periosteum. Development of bone tissues from medullary canal of adjacent bone fragments was also achieved. The newly formed bone was observed on the implantation site in one month after implantation. Implant material (PCL) allowed to assess new bone development within the implant. In this experimental study the highly porous biodegradable construct was stabilized with Ilizarov frame.

We suppose that PCL osteoinductive biodegradable implants coated with HA allow to overcome inconveniences of stress-shielding effect and limited (even absent) radiological assessment typical for metal (titanium alloys and steels) 3D-printed implants. The use of temporary limited external fixation or degradable material providing a long-lasting mechanical resistance represents an option for mechanical resistance while bone healing.

This study did not evaluate the stability of bone fragments when the PCL-HA implant was applied for experimental defect filling. We recognize it as a weak point of the research. Thus, additional studies are required for assessment of biomechanical stability of bone fragments with such type of polymeric implants. Furthermore, morphological and biomechanical studies will clarify the remodeling of bone inside the implant in the process of polymeric matrix degradation.

## 5. Conclusions

In the present study, a method for the restoration of long bone segmental defect with the use of bioactive degradable 3D-printed implant was proposed. Porous biomimetic implant was 3D printed of biodegradable polymer PCL and coated with bioactive mineral HA. Experimental in vivo study with the use of laboratory sheep revealed that the porous degradable 3D implants with bioactive coating located in a long bone segmental defect provide stimulation of reparative osteogenesis and osseointegration into the single implant-bone block. Ingrowth of bone tissue into cylindrical PCL-HA implant from adjacent bone ends and periosteal structures was observed to start from day 7 of study. By the 30th postoperative day, the implant was filled in with bone tissue. Moreover, this type of implant made of degradable polymer allows radiological assessment of reparative osteogenesis in long bone defect filling.

## Figures and Tables

**Figure 1 biomimetics-08-00138-f001:**
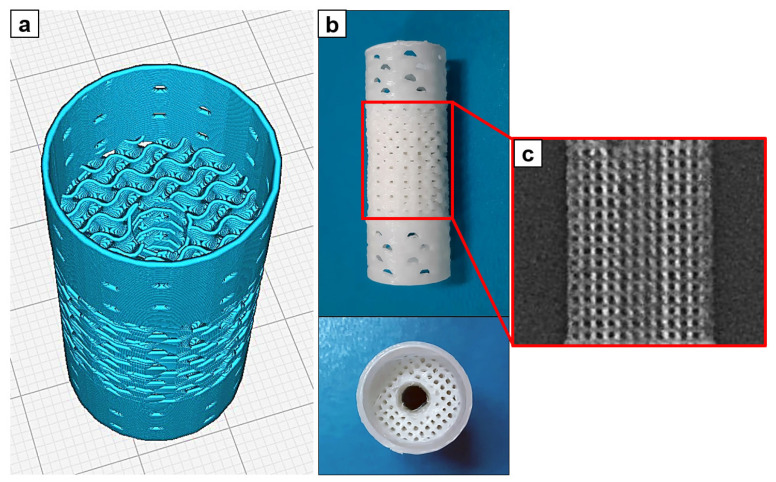
3D-printed cylindrical implant: (**a**)—screenshot of 3D model of the implant from Cura 5.1.0 slicer, (**b**)—3D-printed PCL cylindrical implant; (**c**)—computer tomography of a porous part of the implant (×2.5).

**Figure 2 biomimetics-08-00138-f002:**
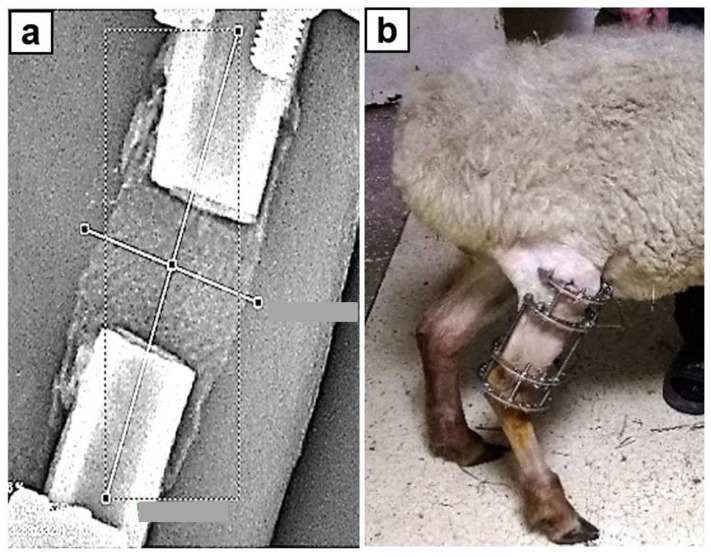
Filling tibia segmental defect with degradable implant: (**a**)—radiograph of a defect filled with the implant (scan lines are marked); (**b**)—photo of an experimental animal on the 4th postoperative day.

**Figure 3 biomimetics-08-00138-f003:**
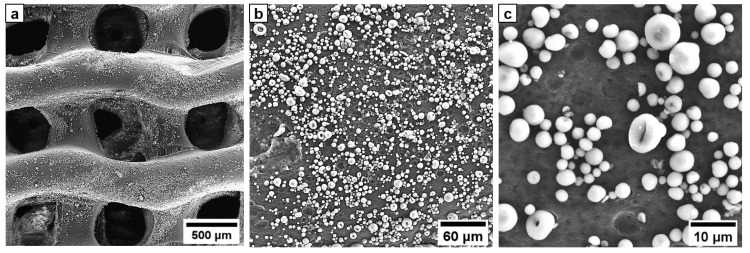
Microscopic appearance of the HA coated gyroid PCL implant: (**a**)—SEM micrograph of PCL implant with HA coating, (**b**)—HA coated surface at 250× magnification, (**c**)—HA coated surface at 1000× magnification.

**Figure 4 biomimetics-08-00138-f004:**
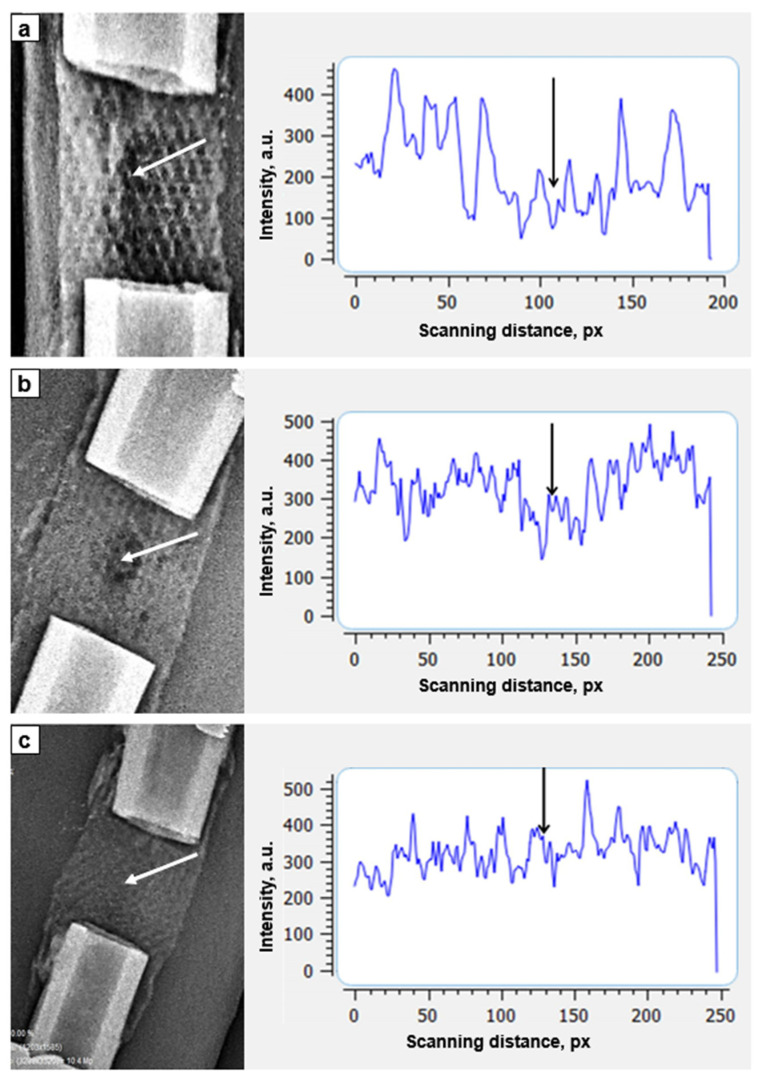
Optical density of the osseointegration zone of an implant (transverse scanning, white arrows indicate the zone of implant axial channel): (**a**)—day of surgery (×1.1); (**b**)—7th days of fixation (×1.0); (**c**)—21st days of fixation (zone of the cortices is expanded, zone of implant axial channel disappeared) (×0.8).

**Figure 5 biomimetics-08-00138-f005:**
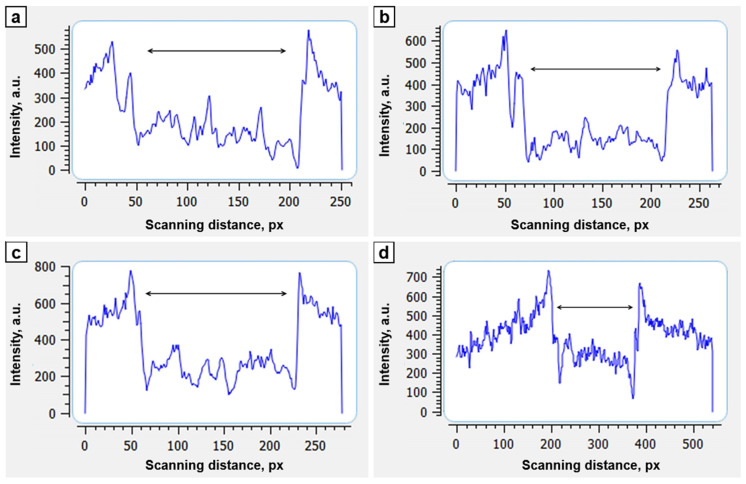
Dynamics of the optical density (longitudinal scanning) of bioactive implant zone (double arrow): (**a**)—the day of the surgery, (**b**)—7th day after surgery, (**c**)—14th day after surgery, (**d**)—28th day after surgery.

**Figure 6 biomimetics-08-00138-f006:**
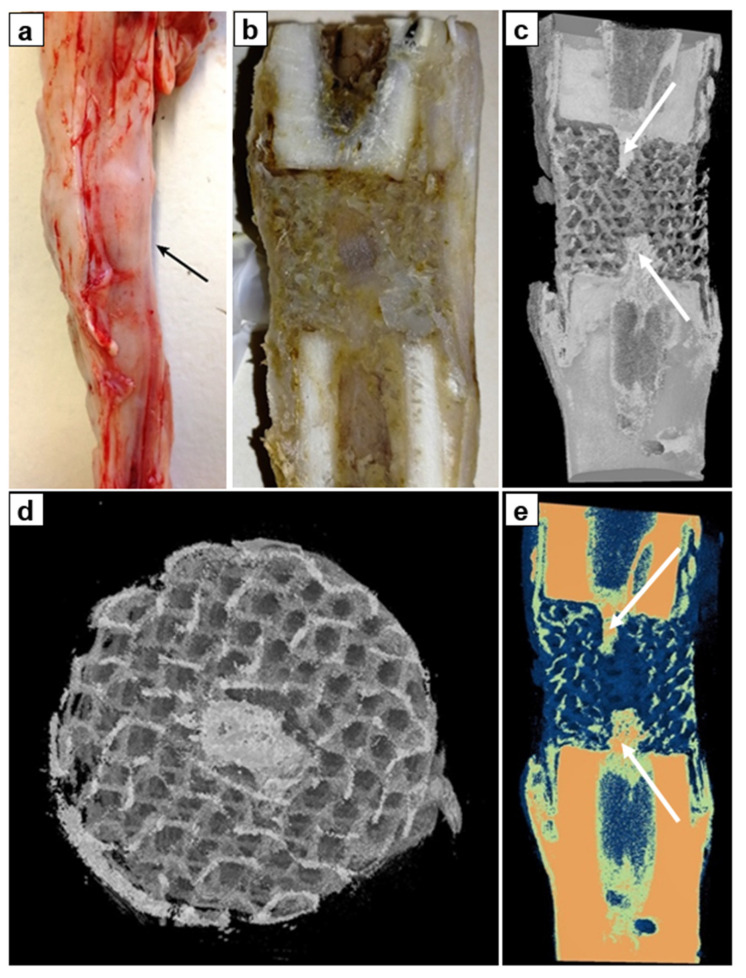
Bone-implant block after 30 days of the experiment: (**a**)—photo of the anatomical preparation of the tibia (the arrow indicates newly formed bone in the central canal), (**b**)—photo of the longitudinal section of the anatomical preparation, (**c**)—CT reconstruction of the anatomical preparation in the range of the shades of gray (×1.1), (**d**)—CT image of a cross-sectioned anatomical preparation in the range of the shades of gray (×2.8), (**e**)—colored visualization of a cross-sectioned anatomical preparation using the Smart colors filter (×1.1).

**Table 1 biomimetics-08-00138-t001:** Dynamics of the optical density of the degradable implant on the bone defect site.

Period of Study, Days	Average Optical Density of the Bone Defect, a.u.	Optical Density of the Shaft of the Adjacent Distal Shaft End, a.u.	Coefficient of Optical Density (Relative to the Distal Bone Fragment)
D0 (surgery)	103 ± 10.7	382 ± 10.4	0.20
7	106 ± 20.4	349 ± 17.4	0.33
14	149 ± 21.2	405 ± 20.1	0.38
21	159 ± 23.1	307 ± 15.5	0.50
28	312 ± 15.9	409 ± 20.3	0.75

## Data Availability

Data are available upon a demand to corresponding author (D.P.).

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
