# Peer review of "Long Bone Defect Filling with Bioactive Degradable 3D-Implant: Experimental Study"

_biomimetics, 2023, doi:10.3390/biomimetics8020138_

Round 1

Reviewer 1 Report

Review on the article Long bone defect filling with bioactive degradable 3D-implant: experimental study, submitted to Biomimetics.

The manuscript presents properly designed research on biodegradable scaffolds used as orthopaedic defect filling. However, some parts are missing certain details, which should have been included. All major and minor critical comments are presented below.

1. There is no justification of the scaffolds’ structure used (pore type, porosity etc.). Why did the authors decided to use gyroid structure? It should be appropriately described, especially as there are several articles presenting more efficient geometries.

2. As the authors printed the implants used it the study, the parameters set for printing should be at least briefly described as their appropriate selection influences the quality of obtained structure, which is crucial in scaffolds.

3. The geometry of the Illizarov fixator should be provided (currently, only parameters of steel wires are included).

4. It is suggested to provide a Figure presenting the design of the used implant.

5. Line 78 contains a typo.

6. It is also suggested to provide another Figure presenting method of implantation.

Author Response

Dear Editors and reviewers of the Biomimetics

We sincerely appreciate the careful review of our manuscript, and the opportunity to answer Reviewer queries. The manuscript has been revised, taking into account the suggestions of the Reviewers.  Below is a point by point response to the Reviewers’ comments.  We thank you for considering the revised manuscript.

Respectfully,

Reviewer 2 Report

In this article, the in vivo behavior of a polymeric porous scaffold within a bone defect is evaluated to assess its osseointegration. The characterization of the material is performed only from a medical point of view and no preliminary and in vitro characterization of this innovative structure was carried out prior to requiring access to animal testing.

- line 62: hydroxyapatite is indicated by the abbreviation HAP, while in the rest of the text with HA.

- What are the reasons for choosing to carry out the study on a number of sheeps equal to five? Being in odd number, how many were of one sex and how many of the other? Why did you choose to use sheeps with age of 1 year? If you had used adult goats, would you have expected different results?

- line 81: it is reported ‘Applied implants for bone defect filling had dimensions of 40 mm height and 18 mm diameter (Figure 1a).’ but later it is said that the height was 20 mm (e.g., line 86). Which one is correct?

- line 88: It is reported that 'Intramedullary channel of 10 mm in diameter was designed at the center of the porous part' but from Figure 1a (top view) the central channel looks very narrow and does not seem to correspond to more than half (55.5%) of the outer diameter (10 mm vs 18 mm).

- line 95: Please specify the instrument set up and extrusion parameters of the PCL (e.g., temperature, extrusion speed, cooling system used…).

- line 97: the protocol used for the coating of hydroxyapatite on the material is completely absent: which organic solvent was used? at what concentration? for how long was the material immersed in such a suspension? Have the effects of this coating (and the action of the solvent) on the material been evaluated?

- line 101: Please specify the acronym ‘AP’.

- line 103: D0 means day 0?

- line 113: it is reported ‘Then cylindrical implant was mounted on the defect site by the integration of free ends of the bone fragments into hollow cylinders on the end face surfaces of the implant’. A schematic representation should be provided. Is it a reproducible method, which therefore allows the comparison of results obtained on different patients/animals?

- line 120: ‘The external frame fixation lasted for 30 days in each animal.’ Is it meant that after 30 days the fixation is broken/damaged? or has it been removed? Why after 30 days?

line 179: Please rephrase ‘Densitometry revealed that all the cells of the implant were filled with the substance which optical density higher than the density of surrounding soft tissues.’ beacause the sentence is not clear.

- line 192: 'was' has been typed twice.

Author Response

(The authors gave the same response as above.)

Round 2

Reviewer 2 Report

-The authors say that 'the Manuscript on the results of our study on the method for coating PCL gyroid scaffolds with HA is already submitted to another journal.' However, all procedures in a scientific article must be well described and reproducible in other laboratories. If any of the protocols and results are to be found in another publication, it should be cited. Otherwise, it is important that you wait for the publication of the preliminary work before submitting this work.

-It is reported (line 77) 'This non-randomized study was performed on five littermate mixed-breed sheep of both sexes aged of 1 year in 2022.', but you said you used only female sheeps. Moreover, goats that young are likely to have a faster bone repair process than adult goats.

-There is still no brief reason why it was decided not to perform in vitro tests before proceeding with those in vivo.

- Tests to assess the delamination of HA coating should be performed.

- Please provide literature to support the choice of using a gyroid geometry.

- The geometry of the Illizarov fixator should be provided.

- It would be highly appreciated if you provide the design of the scaffold fed to the printer, correlated by all measures.

- Why was toluene chosen? What is the effect of immersion of the scaffold in toluene (in which the PCL is soluble)?

- What is the adhesion mechanism of hydroxyapatite on the scaffold? What kind of connection is there between HA and the scaffold?

- Please provide SEM images with larger magnification to evaluate the HA coating, highlighting with an EDX analysis the unequivocal presence of HA.

Author Response

Dear Editors and Reviewer of the Biomimetics

We sincerely appreciate the careful second round review of our manuscript, and the opportunity to answer Reviewer queries. The manuscript has been revised, taking into account the suggestions of the Reviewer.  Please see the attachment of a point by point response to the Reviewer’s comments.  We thank you for considering the revised manuscript.

Respectfully,
